# Organized Breast and Cervical Cancer Screening: Attendance and Determinants in Southern Italy

**DOI:** 10.3390/cancers13071578

**Published:** 2021-03-30

**Authors:** Concetta P. Pelullo, Federica Cantore, Alessandra Lisciotto, Gabriella Di Giuseppe, Maria Pavia

**Affiliations:** Department of Experimental Medicine, University of Campania “Luigi Vanvitelli”, 80138 Naples, Italy; concettapaola.pelullo@unicampania.it (C.P.P.); federica.cantore@unicampania.it (F.C.); alessandralis@hotmail.it (A.L.); gabriella.digiuseppe@unicampania.it (G.D.G.)

**Keywords:** cervical cancer, breast cancer, screening, knowledge, attitudes, attendance

## Abstract

**Simple Summary:**

Although the effectiveness of screening in reducing the mortality of breast and cervical cancer in the EU is established, the impact of these cancers continues to be substantial among women. The aims of this study were to evaluate the attendance to breast and cervical cancer screening and the role of related determinants in an area of Southern Italy. Only 49.7% of the sample reported to have undergone mammography in the previous two years, 27.7% within an organized program and 22% as an opportunistic procedure. The attendance to cervical cancer screening interval of three years was reported by 56.1% of women, 16.1% within an organized program and 40% as an opportunistic procedure. A very low attendance was thus detected for both breast and cervical cancer organized screening programs. A strong commitment to the promotion of these programs is urgently needed, also to support their role in the reduction of inequalities of attendance of disadvantaged women.

**Abstract:**

The aims of this study were to evaluate the attendance to breast and cervical cancer screening and the related determinants in a low attendance area. A cross-sectional study was conducted among mothers of students attending secondary schools and university courses in Campania region, Southern Italy. Only 49.7% of the eligible women reported to have undergone mammography in the previous two years. Unemployed women, unsatisfied about their health status, with a family history of breast cancer, and having visited a physician in the previous 12 months were significantly more likely to have undergone mammography in the previous two years within an organized screening program. The attendance to cervical cancer screening in the interval of three years was reported to be 56.1% of women. Having a lower than graduation degree, being smokers, and having visited a physician in the previous 12 months were significant predictors of having had a Pap-smear in the previous three years in an organized screening program. In this study a very low attendance was found to both breast and cervical cancer organized screening programs. A strong commitment to their promotion is urgently needed, also to reduce inequalities of attendance of disadvantaged women.

## 1. Introduction

Cancer represents the second most common cause of deaths in the European Union (EU), after cardiovascular diseases, and, among cancer primarily affecting women, breast and cervical cancers continue to have a substantial impact [1]. In the EU, the implementation of breast and cervical cancer screening has been effective in reducing the mortality of these diseases [2]. In Italy, breast cancer screening is recommended every 2 years for women aged 50–69 years, and, in the Campania region for women aged 45–69 years [3], whereas cytological screening for cervical cancer is recommended for women from 25 to 64 years of age every 3 years, and a human papillomavirus (HPV) DNA test for women aged 30 to 64 years every 5 years [4]. Organized cervical and breast cancer screening programs have been promoted in Italy since the 90s and are provided at a regional level and implemented by local health units. These programs consist of active invitation of all eligible women at the recommended intervals according to their residence and all the screening, diagnostic and eventually therapeutic pathways are free for all eligible women. All information on organized screening activities is registered and undergoes periodic evaluation according to several quality indicators. In opportunistic screening, instead, women’s attendance is promoted by the recommendation by a healthcare professional or by women’s own choice, it is generally at the women’s own expense, and activities are not registered, so data on its use is not available. Within this context, although primary care physicians (PCP) are not directly involved in the organization and implementation of these programs, nevertheless they play a strategic role in promoting and motivating the participation in screening programs, thanks to their privileged and trusted relationship with patients.

Even if the incidence of invasive cervical cancer and mortality for breast and cervical cancers have been declining in the EU, the different levels of attendance to quality assured screening programs for these cancers produce discrepancies in terms of mortality rates of cervical and breast cancers among the EU countries [2]. In Italy, the introduction of organized screening programs has been characterized by considerable geographical differences [5,6,7,8]. These discrepancies still exist, with Southern regions still showing lower invitation and attendance rates. Indeed, the frequency of invitation in Southern Italy has been reported to be 59% and 74% for breast and cervical cancer screening, respectively [9,10]. Moreover, several determinants that influence attendance to screening programs have been identified, with marked disparities among women in socially or economically disadvantaged groups, as well as in immigrant or ethnic minority populations [11,12,13,14,15]. Furthermore, the coexistence of both opportunistic and organised screening programs makes it difficult to gather comprehensive information about screening coverage [8].

Therefore, the aims of this study were to examine women’s knowledge, and preventive practices about cervical and breast cancer, with specific attention to organized screening programs, in an area of Southern Italy characterised by low attendance to these prevention activities, and to evaluate the determinants of attendance to organized screening as compared to overall attendance including opportunistic breast and cervical cancer screening.

## 2. Materials and Methods

### 2.1. Study Design

This cross-sectional study was conducted from October 2019 to January 2020 among mothers of students attending secondary schools and university courses in the geographic area of Campania region in Southern Italy. The sample was selected through a two-stage cluster sampling technique. First, four secondary schools and three university courses were selected from the list of public secondary schools and university courses of the geographic area. In the second stage, from each university course and school a simple random method was performed to select students whose mothers would be invited to participate in the study.

### 2.2. Data Collection

Each of the participant institutions was contacted by a letter containing the purpose and methodology of the study and the request to conduct the survey in their institution. The data collection tool was a self-administered anonymous structured questionnaire. Two researchers, during regularly scheduled class times, approached the selected students and handed them a sealed envelope containing the questionnaire to be delivered to their mother, as well as the instruction for the returning of the questionnaire. Each mother also received a letter containing information about the rationale and the aims of the study, that completing the questionnaire was voluntary and that the collected information would be processed anonymously and confidentially, and that no personal identifiers were included in the questionnaire. Moreover, it was also explained that data would be analyzed in an aggregate form, and no incentives would be received for participation in the study. This information was also reported on the front page of the questionnaire. Finally, an envelope was provided to return the completed questionnaire. Consent was implied if mothers chose to complete and return the questionnaire.

### 2.3. Questionnaire

The questionnaire was divided into five main sections. The first explored socio-demographic and anamnestic characteristics (age, nationality, education level, marital status, working activity, number of children, chronic diseases, smoking status, alcohol and drug consumption, physical activity, and personal or familiar history of cancer). The second consisted of three questions about knowledge concerning the most frequent cancers in females, cancer screening and cancers that can be detected early. The third examined attitudes towards cancer and cancer prevention, that were measured on 10-points Likert scales. Specifically, the perception of risk of developing cancer was measured with 1 being “not at all worried” and 10 “extremely worried”; beliefs about usefulness of screening tests for cancer prevention with 1 being “not at all useful” and 10 “very useful”; and self-reported health status with responses ranging from 1 for “very unsatisfactory” to 10 for “very satisfactory”. The fourth section explored participants’ practice regarding cancer screening. Responders were asked to indicate whether or not they had visited a physician in the previous 12 months and the related reasons and whether or not they had undergone screening for breast and cervical cancer according to the guideline intervals. Moreover, for each cancer screening women were asked if they had participated in organized or opportunistic screening, time since last breast and cervical screening attendance, and eventual reasons for not having undergone it. Specifically, age eligible women were asked whether they had ever had a Pap-smear or a mammography with possible answers for “no”, “yes, for control” and “yes, because of health problems”; then to those who had had a screening test (for control), time since the last screening test was asked; finally these women were asked whether attendance was preceded by the invitation and appointment by the local health unit, and to those who were not invited, reasons for attendance were asked (recommendation by a physician, own initiative, etc.). The fifth section explored respondents’ source of information about cancer prevention and the need of additional information. A copy of the questionnaire is provided in the Appendix A (Appendix A: Questionnaire). A pilot study was performed on a sample of 59 women, to ensure the readability and clarity of the items. The study protocol was reviewed and approved by the Ethics Committee of the Teaching Hospital of the University of Campania “Luigi Vanvitelli”.

### 2.4. Statistical Analysis

The statistical software Stata (Version 15, StataCorp LLC, College Station, TX, USA) was used to perform the analysis [16]. Descriptive analysis was used to explore the characteristics of the study population. Then, appropriate statistical tests (chi-square, Fisher’s exact and Student’s *t*-test) were conducted in bivariate analysis in order to assess the associations between the independent characteristics and the different outcomes of interest. Following, multivariate stepwise logistic regression analysis was performed to investigate the independent variables associated with five outcomes of interest: accurate knowledge about the most frequent cancers in females and about cancers that can be detected early (no = 0; yes = 1) (Model 1); attendance to Pap-smear screening within organized programs in the previous three years (no = 0; yes = 1) (Model 2); overall attendance to Pap-smear screening in the previous three years (no = 0; yes = 1) (Model 3); attendance to mammography within screening organized programs in the previous two years (no = 0; yes = 1) (Model 4); overall attendance to mammography screening in the previous two years (no = 0; yes = 1) (Model 5). To determine the level of knowledge, an overall knowledge score was constructed considering 1 point for each correct answer to the questions about the most frequent cancers in females, and the cancers that can be detected early. Then, the total knowledge score, ranging from 0 to 15, and the median knowledge score were calculated and respondents who had obtained a score above the median were considered to have accurate knowledge. Socio-demographics and anamnestic characteristics, as well as sources of information about cancer prevention and need of additional information about preventive behavior for cancer from physicians were independent variables included in all models. Moreover, in the models on behaviors related to cancer screening (Models 2–5) independent variables exploring knowledge about cancer and its prevention, attitudes related to cancer and cancer screening, as well as some cancer related behaviors were also included. A detailed description of the independent variables included in each model is reported in a Appendix A (Appendix A: Variables included in the logistic regression models with related categories). The significance levels for the exclusion and inclusion of the variables in the multivariate models were set at 0.4 and 0.2, respectively. Odds ratios (ORs) and 95% confidence intervals (CIs) were calculated.

## 3. Results

### 3.1. Socio-Demographic, Anamnestic and Lifestyle Characteristics

Of the 1088 women invited to participate in the survey, 706 returned the self-administered questionnaire, for a response rate of 64.9%. Participating women s ages ranged from 28 to 67 years. Therefore, they were all eligible for the analysis for cervical cancer screening (25–64), whereas only those 45–67 (418) were included in the analysis for breast cancer screening (45–69). Table 1 provides an overview of the respondents’ socio-demographic, anamnestic, and lifestyle characteristics and related Pap-smear uptake in the previous three years overall and within organized screening programs. Of the survey respondents, 43.2% reported having at least one chronic disease, 55.1% a personal or family history of cancer, 2.6% of cervical and 11.6% of breast cancer. When women were asked about their lifestyle behaviors related to cancer, 21% declared that they were current smokers, 28.2% that they consumed alcohol, and 24.4% reported to practice physical activity. The majority (83.4%) declared that they had visited a physician in the previous 12 months (46.7% a general practitioner, 30.5% a gynecologist, 11.8% another specialist, 4.5% a breast specialist, and 1.9% an oncologist).

### 3.2. Knowledge and Attitudes about Cancer and Related Prevention

When asked about the most frequent cancers in females, almost all (95.6%) correctly identified breast cancer, whereas only 11.8% identified colorectal cancer. Almost all the participants (85.8%) reported that they had heard about cancer screening, 65% from a physician, and 43.2% from the internet. Furthermore, 84.3% knew which cancers can be detected early, and specifically 91.9% and 43% identified breast and cervical as cancers that are detectable early (Table 1).

Overall, only 28.1% of the respondents had an accurate knowledge about the most frequent cancers in females and those that can be detected early, and the results of the multiple logistic regression analysis showed that this knowledge was significantly higher in women with a family history of breast cancer and in those ≥50 years compared to those 45–49 and those ≤44 years (Model 1 in Table 2).

Regarding attitudes, only 17.9% of women were satisfied by their health status and 55.9% considered screening activities effective for cancer prevention. Moreover, more than one-fourth of respondents (26.7%) felt they were at high risk of developing cancer (Table 1).

### 3.3. Cervical Cancer Screening Behavior

Among the eligible women, 603 (85.4%) had ever had a Pap-smear, but only 174 (24.6%) were involved in an organized program, whereas 429 (60.8%) reported an opportunistic procedure. The attendance to the screening interval of three years was reported by 396 (56.1%) of the eligible women, with 114 (16.1%) within an organized program and 282 (40%) as an opportunistic procedure. HPV DNA tests were reported by only 120 (17%) women, and HPV vaccination by 28 (4%). Reported reasons for not having undergone Pap-smears were not having health problems (24.4%), unawareness of being eligible (22%), not having been advised (15.9%), and not having been invited (12.2%).

Women who had undergone a Pap-smear in the preceding three years within an organized screening program were significantly more likely to be unemployed (*p* = 0.024), to have a lower than graduation degree (*p* = 0.017), to have a family history of breast cancer (*p* = 0.031), to be smokers (*p* = 0.022), to know which cancers can be detected early (*p* = 0.026), and to have visited a physician in the previous 12 months (*p* = 0.007) (Table 1). The univariate analysis on attendance to both opportunistic and organized cervical cancer screening showed a different scenario, since attendance was significantly more common in graduated women (*p* = 0.001), with personal or family history of cancer (*p* = 0.001), and of breast cancer (*p* = 0.004), who were physically active (*p* = 0.001), that knew that some cancers may be detected early (*p <* 0.001), and which cancers may be detected early (*p* = 0.003), that believed screening tests are effective (*p* = 0.048), that had visited a physician in the previous 12 months (*p <* 0.001), and had been informed by a physician about cancer prevention (*p <* 0.001) (Table 1).

In the multivariate analysis, only having a lower than graduation degree, being smokers and having visited a physician in the previous 12 months persisted to be associated to having had a Pap-smear in the previous three years in an organized screening program (Model 2 in Table 2), whereas those that predicted getting a Pap-smear in both opportunistic and organized cervical cancer screening were having visited a physician in the previous 12 months, having been informed about cancer screening by a physician and being physically active (Model 3 in Table 2).

### 3.4. Breast Cancer Screening Behavior

Overall, 327 (78.2%) of the 418 eligible women ever reported having undergone a mammography for screening purposes, 166 (39.7%) within an organized program and 161 (38.5%) as an opportunistic procedure. When asked about attendance to mammography in the previous two years, only 208 (49.7%) reported to have undergone it, 116 (27.7%) within an organized program and 92 (22%) as an opportunistic procedure. Among women who had not received mammography, the most frequent reported reasons were not having received an invitation (23.1%), fear of discovering a disease (17.6%), unawareness of being eligible (15.4%), and lack of time (13.2%).

None of the socio-demographic nor the lifestyle characteristics were significantly associated with the correct attendance to an organized mammography screening program, whereas a personal or family history of cancer (*p* = 0.006) or of breast cancer (*p <* 0.001), as well as the correct knowledge that some cancers can be detected early (*p* = 0.021) and which of them can be detected early (*p* = 0.023) were significant predictors of attendance to organized mammography screening programs in the univariate analysis. Analogously, these women were significantly more likely to have visited a physician in the previous 12 months (*p <* 0.001), to have been informed by a physician about cancer prevention (*p* = 0.032), and to be unsatisfied by their health status (*p* = 0.031) (Table 3). When the univariate analysis involved attendance to both opportunistic and organized screening, most of the associations were confirmed, but further predictors were being employed (*p* = 0.043) and having graduated (*p* = 0.002) (Table 3).

In the multivariate analysis, the only characteristics that were significantly associated with attendance to mammography in the previous two years, within an organized screening program, were being unemployed, dissatisfaction with one’s health status, a family history of breast cancer, and having visited a physician in the previous 12 months (Model 4 in Table 2), whereas those that predicted attendance to mammography in both opportunistic and organized screening were having visited a physician in the previous 12 months and having been informed about cancer screening by a physician (Model 5 in Table 2).

Willingness to undertake mammography at the recommended age, investigated among women not yet eligible for breast cancer prevention screening program, revealed that 92.8% of women would be willing to undergo a mammography if eligible.

### 3.5. Sources of Information

Overall, 94.3% of respondents had received information about cancer prevention, 59.2% from physicians and 42.1% from the internet. Moreover, 66% declared that they felt a need for additional information about cancer prevention, and 53% expressed the preference to receive information from physicians. Finally, only 6.1% declared to have participated in cancer prevention information or education activities.

## 4. Discussion

The study has provided an in-depth evaluation of the attendance to mammography and Pap-smear tests, as well as of the related determinants in a Southern Italian population. The main findings of the study are that a very low attendance to organized breast (27.7%) and cervical (16.1%) cancer screening at the recommended time interval was revealed and that the more disadvantaged women are more likely to take part to these programs. The findings of the study may have implications for a more detailed assessment of the role of organized screening programs for the reduction of inequalities in the attendance to cancer screening programs.

### 4.1. Knowledge and Attitudes about Breast and Cervical Cancers and Related Prevention

Knowledge continues to be unsatisfactory and only experience of cancer seems to be a determinant of accurate knowledge, since older age and family history of cancer were the only factors associated to accurate knowledge on the most frequent cancers in females and those who can be detected early. Moreover, it is unacceptable that 14.2% of respondents declared to have never heard about cancer screening; this finding reveals that although organized programs have been implemented more than 20 years ago, communication and education about their role have not yet reached a consistent portion of the eligible population, demanding the implementation of more effective strategies to inform and motivate women to attend these consolidated preventive programs. Likewise, in a previous study, only 0.9% individuals had accurate knowledge about screening programs, while 6.9% of participants were able to identify the aim of the screening programs. The majority (73.0%) had accurate knowledge of the goal of two of the screening programs, while 12.2% of the sample were able to correctly identify the purpose of one of the screening programs [17]. Moreover, in a study conducted among European women, 26.5% correctly knew about the benefits and harms of mammography, with the lowest proportion being among women in Italy (13.3%). Furthermore, 50.9% of Italian women incorrectly believed that mammography screening prevents breast cancer [18]. These findings underline the need for a stronger commitment to the improvement of population education on cancer prevention and specifically on benefits and harms of cancer screening.

### 4.2. Breast and Cervical Cancer Screening Behavior

European Guidelines for the quality assurance of breast and cervical cancer screening recommend their implementation in the framework of organized population-based programs, since they present well documented advantages [19,20]. However, according to our results, attendance to these programs, as well as to opportunistic screening, is far from reaching the recommended target of at least 70% of the eligible population for breast and of 85% for cervical cancer [4]. The overall attendance at the recommended interval, 49.7% for mammography and 56.1% for Pap smears is of great concern, considering that these screening programs are included in the minimum set of healthcare services [21] that should be provided to all eligible subjects by the Italian National Health Service (NHS). An even more concerning finding is that only almost half of the screening mammograms and only 16% out of 56.1% Pap-smears are delivered within organized screening programs. According to these results the size of opportunistic screening is very relevant, specifically for cervical cancer screening. This is of concern, since organized screening programs are more likely to be attended by the socio-economically disadvantaged women, and low coverage in these programs is associated to health and social inequalities [11]. The role of organized cancer screening programs in the reduction of health inequalities is not only related the early detection of cancers, since screening policies include free access to diagnostic tests and treatment. This has been highlighted by a recent study conducted in Italy, that has shown that the reduction of social inequalities associated to organized breast screening programs is not only related to the increase of early diagnoses but also to the improved access to effective treatments [22]. This is of note, considering that social disparities have been demonstrated to influence all the cancer prevention and care pathway, as well as survival and mortality [23,24], and organized screening program may have the potential to eliminate at least some of the barriers encountered by disadvantaged women.

Comparing attendance to analogous cancer screening investigated in studies performed in other populations and/or other countries should be done very cautiously, considering the high heterogeneity involving populations, methods of recruitment of women and monitoring of programs, as well as cultural and socio-economic specific contexts. Attendance rates are very different according to countries. In Europe Northern and Western countries report higher attendance rates to breast and cervical cancer screening compared to Southern and Eastern ones, with attendance to breast cancer ranging from 37% to 90% [6,25,26,27,28], and from 52.1% to 86% for cervical cancer [6,27,29]. In particular, coverage for breast cancer screening in Europe ranges from 49% (East), 62% (West), 64% (North) to 69% (South) [30]. Moreover, Basu et al. reported that overall, only 5.8% and 11.9% of eligible European women have access to breast and cervical cancer screening, respectively [31]. A North-South gradient has been reported also in Italy, where attendance is significantly lower in Southern regions [9].

Indeed, the analysis of our results has shown that the characteristics of women undergoing screening within organized programs are similar regardless of the specific program (mammogram or Pap-smear), and the finding that unemployed and undergraduate women are more likely to attend these programs confirms their role in the reduction of socio-economic barriers to prevention programs. Several studies have investigated the impact of organized screening programs on the reduction of socioeconomic disparities in the attendance of cancer screening activities, using different approaches. Indeed, the role of mammography screening programs in lessening socioeconomic inequalities in mammography practice was revealed in a study reporting the results of two cross-sectional studies in Switzerland before and after the implementation of an organized screening program [32], and a pooled, cross-sectional time series analysis evaluating secondary data from 17 European countries found that where organized screening programs are available, socioeconomic variables, such as education, income and type of employment or unemployment, are not related to attendance to screening and concluded that organized screening programs may reduce the socioeconomic inequalities in attendance to these preventive interventions [7].

It is worth underlying that in all models investigating attendance to screening programs having visited a physician in the previous year was a determinant of attendance, as well as, for the models investigating overall attendance to cervical and breast cancer screening, having been informed about cancer prevention by a physician, confirming the substantial role of physicians in motivating women to undergo preventive interventions, that has already been reported for breast and cervical cancer screening [7,8], as well as for other preventive interventions [33,34,35,36]. This finding underscores the need for healthcare workers to be prepared to play this fundamental role and to be aware of the most effective ways to communicate this message to eligible women.

Another interesting result of the survey is related to the reasons reported by women for not attending breast and cervical cancer screening. It is unacceptable that women did not even know they were eligible for cancer screening or that they did not need cancer screening in the absence of health problems. Indeed, many women reported also having not been advised nor invited to attend cancer screening programs. These findings demand urgent commitment of physicians and public health services to promote these interventions in eligible women. The substantial role of predisposing factors has been reported in a recent systematic review of qualitative studies aimed at the identification of themes influencing attendance to cancer screening in the UK, that demonstrated that the individuals’ relationship and their trust in the health services, as well as their fear of cancer screening and their experiences of risk are among the reasons that influence their response to a screening invitation [37]. Enabling factors, most of which are related to organizational and financial activities put in place by health authorities to implement screening programs also play a substantial role in facilitating women’s attendance. Indeed, it is unacceptable that almost a quarter (23.1%) and 12.2% of eligible women reported to have not received invitation to breast and cervical cancer screening, respectively. However, these figures are relatively optimistic compared to those presented in the 2019 Report of the Italian National Observatory on screening. Here, a coverage of invitation in Southern Italy of 59% and 74% for breast and cervical cancer screening, respectively, was presented [10]. Thus, 41% and 26% of eligible women were not invited to screening.

### 4.3. Limitations

The results of the study should be interpreted considering some potential limitations. First, the cross-sectional nature of the design inherently suggests caution in the interpretation of associations between determinants and outcomes. Moreover, the voluntary participation in the survey does not allow us to exclude “non-participation” bias, with women more engaged with cancer screening to be more likely to participate to the survey compared to the general population; however, the response rate we achieved (64.9%) is considered satisfactory for self-administered questionnaires [38], and allows us to be confident about the external validity of the study. Moreover, attendance to cancer screening activities was self-reported and was not confirmed by objective data; therefore, recall or “desirability” bias cannot be ruled out. Furthermore, it cannot be excluded that some women were overscreened, having undergone screening tests more often than the recommended interval, since they were only asked time since last screening test, and not the number of screening tests performed within the recommended time interval. However, considering the overall low attendance rate, we believe that this limitation did not have substantially affected the results. Finally, it should be acknowledged that the regional origin of the sample of the respondents is a limitation of the study, but we believe that the results may be generalizable to the Italian Southern regions.

## 5. Conclusions

The study has detected an unacceptably low attendance to both breast and cervical cancer screening in Southern Italy, and particularly to organized screening programs. A strong commitment to the promotion of these programs is urgently needed, and the study findings lend also support to the role of these programs in the reduction of inequalities of attendance of disadvantaged women.

## Figures and Tables

**Table 1 cancers-13-01578-t001:** Respondents’ socio-demographic, anamnestic and lifestyle characteristics and related Pap-smear uptake in the previous three years overall and within organized screening programs (*n* = 706).

Characteristics	Total		Overall Pap-Smear Tests in the Previous Three Years		Pap-Smear Tests within Organized Programs in the Previous Three Years	
	*n*	%	*n*	%	*n*	%
Age group (years)						
28–44	286	40.6	152	53.1	48	16.8
45–49	224	31.8	125	55.8	36	16.1
50–67	194	27.6	117	60.3	29	14.9
			χ^2^ = 2.41, 2 df, *p* = 0.300		χ^2^= 0.289, 2 df, *p* = 0.866	
Employment status						
Yes	384	54.4	228	59.4	51	13.3
No	322	45.6	168	52.2	63	19.6
			χ^2^ = 3.69, 1 df, *p* = 0.055		χ^2^ = 5.11, 1 df, *p* = 0.024	
Education level						
Graduate	220	31.4	144	65.5	25	11.4
Undergraduate	481	68.6	252	52.4	89	18.5
			χ^2^ = 10.48, 1 df, *p* = 0.001		χ^2^ = 5.65, 1 df *p* = 0.017	
Marital status						
Married	624	89.1	353	56.6	101	16.2
Other	76	10.9	41	53.9	13	17.1
			χ^2^ = 0.19, 1 df, *p* = 0.663		χ^2^ = 0.04, 1 df, *p* = 0.838	
Number of children						
One child	103	14.7	63	61.2	18	17.5
More than one child	600	85.3	331	55.2	95	15.8
			χ^2^ = 1.28, 1 df, *p* = 0.257		χ^2^ = 0.18, 1 df, *p* = 0.675	
Personal history of chronic diseases						
Yes	305	43.2	169	55.4	52	17
No	401	56.8	227	56.6	62	15.5
			χ^2^ = 0.10, 1 df, *p* = 0.751		χ^2^ = 0.32, 1 df, *p* = 0.570	
Personal or family history of cancer						
Yes	389	55.1	240	61.7	70	18
No	317	44.9	156	49.2	44	13.9
			χ^2^ = 11.05, 1 df, *p* = 0.001		χ^2^ = 2.18,1 df, *p* = 0.139	
Family history of cervical cancer						
Yes	18	2.6	9	50	3	16.7
No	688	97.4	387	56.3	111	16.1
			χ^2^ = 0.28, 1 df, *p* = 0.598		Fisher’s exact *p* = 1.00	
Family history of breast cancer						
Yes	82	11.6	58	70.7	20	24.4
No	624	88.4	338	54.2	94	15.1
			χ^2^ = 8.07, 1 df, *p* = 0.004		χ^2^ = 4.66, 1 df, *p* = 0.031	
Current smokers						
Yes	148	21	80	54.1	33	22.3
No	558	79	316	56.6	81	14.5
			χ^2^ = 0.32, 1 df, *p* = 0.574		χ^2^ = 5.23, 1 df, *p* = 0.022	
Alcohol consumption						
Yes	199	28.2	130	65.3	35	17.6
No	507	71.8	266	52.5	79	15.6
			χ^2^ = 9.59, 1 df, *p* = 0.002		χ^2^ = 0.42, 1 df, *p* = 0.515	
Physical activity						
Yes	172	24.4	116	67.4	24	14
No	534	75.6	280	52.4	90	16.9
			χ^2^ = 11.89, 1 df, *p* = 0.001		χ^2^ = 0.81, 1 df, *p* = 0.369	
Knowledge that some cancers can be detected early						
Yes	606	85.8	357	58.9	102	16.8
No	100	14.2	39	39	12	12
			χ^2^ = 13.81, 1 df, *p* < 0.001		χ^2^ = 1.48, 1 df, *p* = 0.224	
Knowledge of which cancers can be detected early						
Yes	595	84.3	348	58.5	104	17.5
No	111	15.7	48	43.2	10	9
			χ^2^ = 8.82, 1 df, *p* = 0.003		χ^2^ = 4.95, 1 df, *p* = 0.026	
Accurate knowledge (about the most frequent cancers in females and cancers than can be detected early)						
Yes	198	28.1	126	63.6	34	17.2
No	508	78.9	270	53.1	80	15.7
			χ^2^ = 6.36, 1 df, *p* = 0.012		χ^2^ = 1.05, 1 df, *p* = 0.305	
Perception of personal health status						
Satisfactory	122	17.9	71	58.2	18	14.8
Unsatisfactory	560	82.1	318	56.8	92	16.4
			χ^2^ = 0.08, 1 df, *p* = 0.775		χ^2^ = 0.21, 1 df *p* = 0.649	
Perceived risk of developing cancer						
High	185	26.7	103	55.7	34	18.4
Low	507	73.3	289	57	79	15.6
			χ^2^ = 0.09, 1 df, *p* = 0.755		χ^2^ = 0.77, 1 df, *p* = 0.378	
Perceived effectiveness of screening tests						
High	387	55.9	232	60	67	17.3
Low	305	44.1	160	52.5	47	15.4
			χ^2^ = 3.89, 1 df, *p* = 0.048		χ^2^= 0.45, 1 df, *p* = 0.503	
Having visited a physician in the previous 12 months						
Yes	589	83.4	364	61.8	105	17.8
No	117	16.6	32	27.4	9	7.7
			χ^2^ = 47.03, 1 df, *p <* 0.001		χ^2^ = 7.40, 1 df, *p* = 0.007	
Having been informed about cancer screening by a physician						
Yes	414	68.3	253	61.1	73	17.6
No	192	31.7	104	54.2	29	15.1
			χ^2^ = 2.61, 1 df, *p* = 0.106		χ^2^ = 0.60, 1 df *p* = 0.439	
Having been informed about cancer prevention by physicians						
Yes	418	59.2	270	64.6	72	17.2
No	288	40.8	126	43.8	42	14.6
			χ^2^ = 30.07, 1 df, *p* < 0.001		χ^2^ = 0.88, 1 df, *p* = 0.349	
Need of additional information about cancer prevention from physicians						
Yes	374	53	216	58.2	69	18.6
No	332	47	176	54.8	44	13.7
			χ^2^ = 0.80, 1 df, *p* = 0.369		χ^2^ = 3.01, 1 df, *p* = 0.083	

Number for each item may not add up to total number of study population due to missing values.

**Table 2 cancers-13-01578-t002:** Logistic regression models results.

**Model 1. Accurate Knowledge (about the Most Frequent Cancers in Females and Cancers that could be Detected Early)**	**OR ***	**SE ^+^**	**95% CI ^°^**	***p*-Value**
Log likelihood= −391.74; χ^2^= 38.28 (7 df); *p* < 0.0001				
Age (years)				
28–44	0.6	0.13	0.39–0.93	0.021
45–49	0.63	0.14	0.41–0.98	0.039
50–67	1.00 ^α^	-	-	-
Family history of breast cancer				
No	1.00 ^α^	-	-	-
Yes	1.67	0.42	1.01–2.74	0.044
Education level				
Undergraduate	1.00 ^α^	-	-	-
Graduate	1.45	0.3	0.96–2.18	0.079
Employment status				
No	1.00 ^α^	-	-	-
Yes	1.37	0.28	0.92–2.03	0.121
Personal history of chronic diseases				
No	1.00 ^α^	-	-	-
Yes	0.74	0.13	0.52–1.05	0.088
Having been informed about cancer prevention by physicians				
No	1.00 ^α^	-	-	-
Yes	1.42	0.27	0.98–2.06	0.066
**Model 2. Attendance to Pap-Smear within Organized Programs in the Previous Three Years**	**OR ***	**SE ^+^**	**95% CI ^°^**	***p*-Value**
Log likelihood= −277.83; χ^2^= 33.31 (8 df); *p* < 0.0001				
Having visited a physician in the previous 12 months				
No	1.00 ^α^	-	-	-
Yes	2.96	1.23	1.31–6.67	0.009
Education level				
Undergraduate	1.00 ^α^	-	-	-
Graduate	0.54	0.16	0.31–0.96	0.035
Current smokers				
No	1.00^α^	-	-	-
Yes	1.71	0.42	1.05- 2.78	0.03
Family history of breast cancer				
No	1.00 ^α^	-	-	-
Yes	1.76	0.54	0.96–3.22	0.067
Employment status				
No	1.00 ^α^	-	-	-
Yes	0.71	0.17	0.44–1.15	0.166
Knowledge that some cancers can be detected early				
No	1.00 ^α^	-	-	-
Yes	1.74	0.64	0.85–3.57	0.132
Need of additional information about cancer prevention from physicians				
No	1.00 ^α^	-	-	-
Yes	1.4	0.31	0.91–2.17	0.127
Having been informed about cancer prevention by a physician				
No	1.00 ^α^	-	-	-
Yes	1.28	0.31	0.8–2.05	0.305
**Model 3. Overall Attendance to Pap-Smear in the Previous Three Years**	**OR ***	**SE ^+^**	**95% CI ^°^**	***p*-Value**
Log likelihood= −410.13; χ^2^= 83.5 (8 df); *p* < 0.0001				
Having been informed about cancer prevention by physicians				
No	1.00 ^α^	-	-	-
Yes	1.78	0.32	1.25–2.53	0.001
Having visited a physician in the previous 12 months				
No	1.00 ^α^	-	-	-
Yes	3.56	0.86	2.21–5.73	<0.001
Physical activity				
No	1.00 ^α^	-	-	-
Yes	1.66	0.33	1.11–2.46	0.013
Perceived effectiveness of screening tests, continuous	1.09	0.05	0.99–1.19	0.059
Education level				
Undergraduate	1.00 ^α^	-	-	-
Graduate	1.34	0.26	0.92–1.96	0.122
Knowledge that some cancers can be detected early				
No	1.00 ^α^	-	-	-
Yes	1.24	0.3	0.77–2.01	0.381
Alcohol consumption				
No	1.00 ^α^	-	-	-
Yes	1.29	0.24	0.89–1.87	0.176
Family history of breast cancer				
No	1.00 ^α^	-	-	-
Yes	1.45	0.4	0.84–2.51	0.177
**Model 4. Attendance to Mammography within Organized Programs in the Previous Two Years**	**OR ***	**SE ^+^**	**95% CI ^°^**	***p*-Value**
Log likelihood= −207.6; χ2= 49.31 (8 df); *p* < 0.0001				
Family history of breast cancer				
No	1.00 ^α^	-	-	-
Yes	2.84	0.94	1.49–5.43	0.002
Employment status				
No	1.00 ^α^	-	-	-
Yes	0.52	0.13	0.32–0.86	0.010
Perception of personal health status				
Unsatisfactory	1.00 ^α^	-	-	-
Satisfactory	0.46	0.17	0.22–0.94	0.034
Having visited a physician in the previous 12 months				
No	1.00 ^α^	-	-	-
Yes	7.78	4.79	2.33–25.98	0.001
Age (years)				
45–49	0.72	0.18	0.45–1.17	0.185
50–67	1.00 ^α^	-	-	-
Knowledge that some cancers can be detected early				
No	1.00 ^α^	-	-	-
Yes	2.11	0.97	0.86–5.19	0.104
Perceived risk of developing cancer, continuous	1.08	0.06	0.98–1.2	0.126
Perceived effectiveness of screening tests, continuous	1.07	0.08	0.92–1.24	0.395
**Model 5. Overall Attendance to Mammography in the Previous Two Years**	**OR ***	**SE ^+^**	**95% CI ^°^**	***p*-Value**
Log likelihood= −240.95; χ2= 55.82 (7 df); *p* < 0.0001				
Having visited a physician in the previous 12 months				
No	1.00 ^α^	-	-	-
Yes	3.99	1.53	1.88–8.47	<0.001
Having been informed about cancer prevention by physicians				
No	1.00 ^α^	-	-	-
Yes	2.1	0.5	1.31–3.36	0.002
Perceived risk of developing cancer, continuous	1.09	0.05	0.99–1.19	0.064
Education level				
Undergraduate	1.00 ^α^	-	-	-
Graduate	1.5	0.35	0.95–2.36	0.081
Perceived effectiveness of screening tests, continuous	1.11	0.07	0.98–1.27	0.094
Family history of breast cancer				
No	1.00 ^α^	-	-	-
Yes	1.75	0.61	0.88- 3.48	0.108
Alcohol consumption				
No	1.00^α^	-	-	-
Yes	1.29	0.31	0.8–2.07	0.292

* Odds Ratio; ^+^ Standard Error; ^°^ Confidence Interval; ^α^ Reference category; Variables deleted by backward elimination procedure were not included in the table.

**Table 3 cancers-13-01578-t003:** Respondents’ socio-demographic, anamnestic and lifestyle characteristics and related attendance to mammography in the previous two years overall and within organized screening programs (*n* = 418).

Characteristics	Total		Overall Attendance to Mammography in the Previous Two Years		Attendance to Mammography within Organized Programs in the Previous Two Years	
	*n*	%	*n*	%	*n*	%
Age group (years)						
28–44			-	-	-	-
45–49	224	53.6	104	46.4	54	24.1
50–67	194	46.4	104	53.6	62	32
			χ^2^ = 2.14, 1 df, *p* = 0.143		χ^2^ = 3.19, 1 df, *p* = 0.074	
Employment status						
Yes	253	60.5	136	53.8	63	24.9
No	165	39.5	72	43.6	53	32.1
			χ^2^ = 4.09, 1 df, *p* = 0.043		χ^2^ = 2.59, 1 df, *p* = 0.107	
Education level						
Graduate	165	39.7	98	59.4	47	28.5
Undergraduate	251	60.3	110	43.8	69	27.5
			χ^2^ = 9.65, 1 df, *p* = 0.002		χ^2^ = 0.04, 1 df, *p* = 0.825	
Marital status						
Married	358	86.5	185	51.7	105	29.3
Other	56	13.5	22	39.2	10	17.9
			χ^2^ = 2.97, 1 df, *p* = 0.085		χ^2^ = 3.17, 1 df, *p* = 0.075	
Number of children						
One child	68	16.3	35	51.5	16	23.5
More than one child	348	83.7	173	49.7	100	28.7
			χ^2^ = 0.07, 1 df, *p* = 0.791		χ^2^ = 0.76, 1 df, *p* = 0.381	
Personal history of chronic diseases						
Yes	194	46.4	98	50.5	56	28.9
No	224	53.6	110	49.1	60	26.8
			χ^2^ = 0.08, 1 df, *p* = 0.774		χ^2^ = 0.22, 1 df, *p* = 0.636	
Personal or family history of cancer						
Yes	247	59.1	139	56.2	81	32.8
No	171	40.9	69	40.4	35	20.5
			χ^2^ = 10.24, 1 df, *p* = 0.001		χ^2^ = 7.65, 1 df, *p* = 0.006	
Family history of cervical cancer						
Yes	11	2.6	6	54.5	4	36.4
No	407	97.4	202	49.6	112	27.5
			χ^2^ = 0.10, 1 df, *p* = 0.748		Fisher’s exact *p* = 0.507	
Family history of breast cancer						
Yes	53	12.7	37	69.8	27	50.9
No	365	80.4	171	46.8	89	24.4
			χ^2^ = 9.76, 1 df, *p* = 0.002		χ^2^ = 16.28, 1 df, *p* < 0.001	
Current smokers						
Yes	82	19.6	35	42.7	21	25.6
No	336	80.4	173	51.5	95	28.3
			χ^2^ = 2.04, 1 df, *p* = 0.153		χ^2^ = 0.23, 1 df, *p* = 0.629	
Alcohol consumption						
Yes	123	29.4	74	60.2	40	32.5
No	295	70.6	134	45.4	76	25.8
			χ^2^ = 7.54, 1 df, *p* = 0.006		χ^2^ = 1.97, 1 df, *p* = 0.160	
Physical activity						
Yes	111	25.6	59	53.2	29	26.1
No	307	73.4	149	48.5	87	28.3
			χ^2^ = 0.69, 1 df, *p* = 0.404		χ^2^ = 0.19, 1 df, *p* = 0.655	
Knowledge that some cancers can be detected early						
Yes	368	88	192	52.2	109	29.6
No	50	12	16	32	7	14
			χ^2^ = 7.16, 1 df, *p* = 0.007		χ^2^ = 5.35, 1 df, *p* = 0.021	
Knowledge of which cancers can be detected early						
Yes	364	87.1	191	52.5	108	29.7
No	54	12.9	17	31.5	8	14.8
			χ^2^ = 8.28, 1 df, *p* = 0.004		χ^2^ = 5.17, 1 df, *p* = 0.023	
Accurate knowledge (about the most frequent cancers in females and cancers than can be detected early)						
Yes	131	31.3	71	54.2	36	27.5
No	287	68.7	137	47.7	80	27.9
			χ^2^ = 1.50, 1 df, *p* = 0.220		χ^2^ = 0.01, 1 df, *p* = 0.934	
Perception of personal health status						
Satisfactory	68	17	34	50	12	17.7
Unsatisfactory	333	83	168	50.5	102	30.6
			χ^2^ = 0.01, 1 df, *p* = 0.946		χ^2^ = 4.67, 1 df, *p* = 0.031	
Perceived risk of developing cancer						
High	99	24.4	55	55.6	33	33.3
Low	307	75.6	149	48.5	80	26
			χ^2^ = 1.47, 1 df, *p* = 0.224		χ^2^ = 1.97, 1 df, *p* = 0.160	
Perceived effectiveness of screening tests						
High	228	55.7	124	54.4	72	31.6
Low	181	44.3	83	45.9	43	23.8
			χ^2^ = 2.93, 1 df, *p* = 0.087		χ^2^ = 3.05, 1 df, *p* = 0.081	
Having visited a physician in the previous 12 months						
Yes	362	86.6	198	54.7	113	31.2
No	56	13.4	10	17.9	3	5.4
			χ^2^ = 26.32, 1 df, *p* < 0.001		Fisher’s exact *p* < 0.001	
Having been informed about cancer screening by a physician						
Yes	270	73.4	146	54	84	31.1
No	88	26.6	46	46.9	25	25.5
			χ^2^ = 1.46, 1 df, *p* = 0.226		χ^2^ = 1.08, 1 df, *p* = 0.298	
Having been informed about cancer prevention by physicians						
Yes	265	63.4	157	59.3	83	31.2
No	153	36.6	51	33.3	33	21.6
			χ^2^ = 26.05, 1 df, *p* < 0.001		χ^2^ = 4.60, 1 df, *p* = 0.032	
Need of additional information about cancer prevention from physicians						
Yes	213	51	106	49.8	59	27.7
No	205	49	102	49.8	57	27.8
			χ^2^ = 0.01, 1 df, *p* = 0.981		χ^2^ = 0.01, 1 df, *p* = 0.999	

Number for each item may not add up to total number of study population due to missing values.

## Data Availability

The data presented in this study are available on request from the corresponding author.

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
