# Peer review of "Organized Breast and Cervical Cancer Screening: Attendance and Determinants in Southern Italy"

_cancers, 2021, doi:10.3390/cancers13071578_

Round 1

Reviewer 1 Report

It is the purpose of this paper to find determinants of low participation in cervical and breast screening in an area in Southern Italy. It is a survey based on self-administered questionnaires. The sampling of women asked to participate is somewhat untraditional, but OK given that it was probably the best possibility. 

The reporting is unacceptable. 

  1. all the attitude questions are not really so information, I would skip all of them
  2. Table 1 can be merged with the first page of Table 4 and Table5 (up to including physical activity) - then one will be able to see how percentages are calculated, etc.
  3. All other tables can be deleted
  4. Description of the screening programme is missing: how is the organized programme: invitations, payment, etc. How is the opportunistic screening: payment, etc.
  5. First paragraph of Discussion should clearly state what was the main finding

The results of this small survey is potentially interesting, but as it is now too many uninteresting numbers and too little context is provided to make this paper interesting for a wider audience. 

Author Response

Reviewer 1:

  • all the attitude questions are not really so information, I would skip all of them

[Lines 212-215] In response to this point, we have investigated attitudes with two questions (See questionnaire), and the reason is related to their potential influence on screening attendance. According to your suggestion, we have now eliminated Table 2 where attitudes were described, reduced their description to a paragraph that follows the description of knowledge, and eliminated the model on attitudes, and its description in the results and in the discussion. Their role as potential determinants of attendance to cervical and breast cancer screening is shown in the Tables 4 and 5 that in the “new” version are Tables 1 and 3, and as independent variables of the logistic regression models (Table 2).

  • Table 1 can be merged with the first page of Table 4 and Table5 (up to including physical activity) - then one will be able to see how percentages are calculated, etc.

As suggested, we have merged Table 1 and 2 with Table 4 and 5, and now they are Tables 1 and 3.

  • All other tables can be deleted

As suggested, we have deleted Table 1 and 2.

  • Description of the screening programme is missing: how is the organized programme: invitations, payment, etc. How is the opportunistic screening: payment, etc.

[Lines 44-53] As suggested, description of organized and opportunistic screening has been provided in the Introduction.

  1. First paragraph of Discussion should clearly state what was the main finding.

As suggested, in the Discussion section we have added the main finding of the study.

Reviewer 2 Report

Measuring attendance in cancer screening programs and determinants thereof may help identify opportunities to develop interventions and ways to improve programs that reduce disparities. For that reason, your manuscript is very relevant for policy makers in this southern Italian region!

General remarks:

  • You use serval terms (attendance, participation, uptake, adherence) as synonyms for each other. Without a proper definition, this un-sharp writing can be misleading.
  • The manuscript (or the supplementary materials) is lacking the full questionnaire. The paper would be so much more readable with the complete questions at hand!

Introduction

Briefly mention when the screening programs in Italy were initiated and some quality indicators (e.g. invitation coverage), the fact that is it free of charge etc. A bit more context is very helpful in understanding the later analyses, the choice for specific questions like the one regarding visits from the physician etc.

As aim of your study, you state the evaluation of determinants of access to screening. However, I believe you rather look at determinants of test utilization to early detection of cancer. Determinants of access to screening would be practical issues or organisational barriers such as limited capacity of screening units, personnel etc.  I suggest rephrasing the aim of your study.

Materials and Methods – Questionnaire

I suggest adding the original questionnaire to the supplementary materials. It is otherwise very difficult to judge the precision of the questions. Specifically, the fourth section of the questionnaire seems relevant to the aim of the paper, but the exact questions are not comprehensible. Are the women able to distinguish between a true screening mammography (non symptomatic) from a clinical mammography due to symptoms. Can a distinction really be made between population-based screening by invitation, opportunistic testing or even the use of a test because of symptoms or other medical reason? This distinction is necessary in order to conclude reasons behind cancer testing.

One of the core questions in your analysis seems to be the one regarding have been visited by a physician. As non-Italian, I do not understand the relevance of this question. Is it the norm to receive a home visit by a doctor of GP? Which role does that play in the context of general health care and in the context of early detection of breast or cervical cancer?

Results

40.6% of the interviewed women are younger than 44 years, thus not part of the target age range for breast cancer screening in Campania. I wonder if these women were excluded from breast cancer screening related analysis, as they are not relevant and might even confound the results. The same goes for women “≥50”: can you make sure no women older than 69 are included?

Table 1 seems to be missing a footnote. Some variables in the first column are marked with ** but I cannot find a description of their meaning. Why are not all characteristics presented in table1? Nationality, drug consumption, dietary habits are missing.

Table 2 seems to be lacking some of the items as well. In the text, you mention that knowledge regarding cancers other than breast and cervical was asked as well. Again, without a full questionnaire it is hard to comprehend. And the presentation of results ends up looking arbitrary (or selective, which would be worse of course).

In the text below Table 2, you mention that 28.1% of the respondents had accurate knowledge about most frequent cancers in women. Again, I have no idea where to look up this number in any of the tables.

Model 3 Table 3: are the numbers presented correctly? On the bottom of page 6, you present the numbers of the logistic regression of mammography uptake. Here it looks as if being employed (YES) leads to a significant drop in uptake while on page 11 you describe the results the other way around (OR of 0.52 for unemployed women). That is very confusing. Moreover, you call it “adherence to mammography”. In my understanding, in screening research you call adherence the willingness to follow an invitation. Therefore, these results should be only reported for those women who received an invitation for the organised program. Or you stick to calling it “uptake”. But I suggest to remain very clear about what you mean and try to avoid different terms for the same thing.  

At the bottom of page 8, you report the results on breast cancer screening behaviour. I wonder, again, whether or not you excluded women outside the target age range for organised screening on Campania when you look at reasons for reasons to not be screened. Not having been invited seems to be most relevant. However, of course only for women 45-69. Potential (organisational, financial) barriers towards a complete invitational coverage of eligible women should definitely be included in the discussion of this paper!

@Source of information, section 3.7: Without the complete questionnaire at hand, it is impossible to know what the women could choose from in term of source of information about cancer prevention. Women invited to screening should have received information about not only the screening procedure but also about the usefulness, effectiveness, harms and benefits of screening. Is that sufficient information?

Discussion

Needs editing: first word out of context. Generally, sentences are too long and winded and could easily be separated in order to increase readability, e.g. first sentence, split after “organized screening programs, …”

4.1. Starts with a statement regarding knowledge on cancer screening. I suggest adding a brief review of relevant research (results) to increase context.  

Last paragraph of page 14, “furtherly confirmed” should be edited.

First paragraph page 15, edit sentence starting with “As expected…”It is too long and the reader loses the meaning of the results. Also, re-write “higher knowledge” into “accurate knowledge”.

4.2.  Do you also consider the possibility that women are overscreened in terms of more often than the recommended 24 months interval (for breast cancer)? Without the questionnaire at hand, I do not know the exact question you asked and whether there is the possibility of a sub-group of women to have been invited to the screening program PLUS attending opportunistic screening. There are two ends from the “recommended interval”.
Please edit and shorten the sentence starting with “According to these results,…”. The information gets lost.

I suggest including different publications to refer to participation rate differences across Europe, e.g. the European screening report (Basu P et al., Int J Cancer 2018) and Zielonke et.al. (Int J Cancer, 2020), who also collected examination coverage of organised as well as opportunistic breast cancer screening.

In the last paragraph of page 15, you start using different terms for the same thing, although there clearly are differences between “inequality” and “inequity”. Make sure you use these terms coherently!

In Table 2 you present that a shockingly high 14.2% of respondent claim to have never heard about cancer screening. THAT is a result to hand over to the responsible stakeholders! Which leads me to the next comment: which stakeholders other than GP are responsible for a successful screening program? The paper would really benefits if the scope of the discussion would be widened to give the reader more context.

This study focused on socio-economic inequality in the use of cancer testing. However, inequality can be present in each phase along the cancer care pathway, so inequalities in screening findings, survival, and in the access to cancer treatment could also be taken into account. Furthermore, deprived women with screen-detected breast cancer are more likely to face barriers like travel time and distance, inability to take time off work, and lack of health information (Bigby J, Cancer Causes Control 2005; Bradley CJ et al., J Natl Cancer Inst 2002). An Italian study found that the breast cancer screening programme reduced disparities in the access to treatment (Zengarini N, Eur J Cancer Prev 2016). I suggest extending the discussion.

Conclusions

Edit first line “unacceptably low”

Author Response

 Reviewer 2

General remarks:

  • You use serval terms (attendance, participation, uptake, adherence) as synonyms for each other. Without a proper definition, this un-sharp writing can be misleading.

As suggested, we have standardized the terms used in the text, using only the term “attendance” that better explain the aim of the study.

  • The manuscript (or the supplementary materials) is lacking the full questionnaire. The paper would be so much more readable with the complete questions at hand!

As suggested, we have added as Supplementary Material (S1) the English version of the questionnaire used in the survey.

Introduction

- Briefly mention when the screening programs in Italy were initiated and some quality indicators (e.g. invitation coverage), the fact that is it free of charge etc. A bit more context is very helpful in understanding the later analyses, the choice for specific questions like the one regarding visits from the physician etc.

[Lines 53-57] As suggested, in the Introduction section we have added further information about screening programme in Italy, as well as about the role of the General Practitioner.

- As aim of your study, you state the evaluation of determinants of access to screening. However, I believe you rather look at determinants of test utilization to early detection of cancer. Determinants of access to screening would be practical issues or organisational barriers such as limited capacity of screening units, personnel etc. I suggest rephrasing the aim of your study.

As suggested, we have changed the aim of our study from “determinants of access to screening” in “determinants of attendance to screening”.

Materials and Methods – Questionnaire

- I suggest adding the original questionnaire to the supplementary materials. It is otherwise very difficult to judge the precision of the questions. Specifically, the fourth section of the questionnaire seems relevant to the aim of the paper, but the exact questions are not comprehensible. Are the women able to distinguish between a true screening mammography (non-symptomatic) from a clinical mammography due to symptoms. Can a distinction really be made between population-based screening by invitation, opportunistic testing or even the use of a test because of symptoms or other medical reason? This distinction is necessary in order to conclude reasons behind cancer testing.

As suggested, we have added the English version of the questionnaire used in this study, in which it is possible to distinguish between a true screening mammography (for control) from a clinical mammography due to symptoms. In the method section we have added the questions that were used to distinguish mammography for screening from that for diagnostic purposes, as well as to distinguish opportunistic from organized screening.

- One of the core questions in your analysis seems to be the one regarding have been visited by a physician. As non-Italian, I do not understand the relevance of this question. Is it the norm to receive a home visit by a doctor of GP? Which role does that play in the context of general health care and in the context of early detection of breast or cervical cancer?

[Lines 122-127] In response to this point, in Italy every citizen chooses a personal primary care physician (PCP), that is the first figure to refer to for each health need, representing the gatekeeper for all other healthcare services, and it has been demonstrated that patients trust and rely on PCP to receive advice and support for any decision on their health. Therefore, the PCP, although not directly involved in the organization of screening, nevertheless has a strategic role in encouraging the participation of citizens in screening programs, thanks to the privileged and direct relationship. Based on this, our hypothesis was that having been in touch with the PCP would have been an opportunity to be motivated to participate in screening programs. This has now been included in the Introduction.

Results

- 40.6% of the interviewed women are younger than 44 years, thus not part of the target age range for breast cancer screening in Campania. I wonder if these women were excluded from breast cancer screening related analysis, as they are not relevant and might even confound the results. The same goes for women “≥50”: can you make sure no women older than 69 are included?

[Lines 172-175] In response to this point, participating women’s age ranged from 28 to 67 years. Therefore, they were all eligible for the analysis for cervical cancer screening (25-64), whereas only those 45-67 (418) were included in the analysis for breast cancer screening (45-69). We included women until 68 year since they could have had undergone Pap test in the previous three years. This has now specified in the results section.

- Table 1 seems to be missing a footnote. Some variables in the first column are marked with ** but I cannot find a description of their meaning. Why are not all characteristics presented in table1? Nationality, drug consumption, dietary habits are missing.

Thanks for your note. Indeed, footnote on Table 1 has been erroneously cancelled during editing. However, according to Reviewer 1 suggestion, we have eliminated Table 1 and the related information is now available in the “new” Table 1 and the “new” Table 3, referring to the population eligible for cervical and breast cancer screening, respectively.

As regard to nationality and drug consumption they were not included in the Results, since women were all Italians and none of them declared to have used drugs. As regards to dietary habits, it was a typo in the methods, since we did not ask about it in the survey. Therefore, we have now eliminated this mistake from the methods section.

- Table 2 seems to be lacking some of the items as well. In the text, you mention that knowledge regarding cancers other than breast and cervical was asked as well. Again, without a full questionnaire it is hard to comprehend. And the presentation of results ends up looking arbitrary (or selective, which would be worse of course).

In Table 2 we reported the results related to knowledge and attitudes on cancer and screening in general and then specifically those related to breast and cervical cancer. Now, according to Reviewer 1, this information is reported in “new” Tables 1 and 3. As suggested, in Supplementary Material (S1) we have added the Questionnaire, where the formulation of questions on what kind of knowledge was retrieved is displayed.

- In the text below Table 2, you mention that 28.1% of the respondents had accurate knowledge about most frequent cancers in women. Again, I have no idea where to look up this number in any of the tables.

[Lines 149-155] In response to this point, the results on “Accurate knowledge” is now reported in Tables 1 and 3, and the way this variable was constructed is reported in the methods section, as follows: “To determine the level of knowledge, an overall knowledge score was constructed considering 1 point for each correct answer to the questions about the most frequent cancers in females, and the cancers that can be early detected. Then, the total knowledge score, ranging from 0 to 15, and the median knowledge score were calculated and respondents who had obtained a score above the median were considered to have accurate knowledge.”

- Model 3 Table 3: are the numbers presented correctly? On the bottom of page 6, you present the numbers of the logistic regression of mammography uptake. Here it looks as if being employed (YES) leads to a significant drop in uptake while on page 11 you describe the results the other way around (OR of 0.52 for unemployed women). That is very confusing.

In response to this point, we confirm that the OR of 0.52 reported in Model and in the results on page 11 refers to the probability of employed compared to unemployed women to attend organized breast cancer screening programs, thus showing that employed women are less likely to attend compared to unemployed, that is the same as saying that unemployed are more likely than employed to attend, and this is the way it has been reported in the results. To avoid misunderstanding, we have eliminated all the OR from the text, since they are already reported in the Tables.

- Moreover, you call it “adherence to mammography”. In my understanding, in screening research you call adherence the willingness to follow an invitation. Therefore, these results should be only reported for those women who received an invitation for the organised program. Or you stick to calling it “uptake”. But I suggest to remain very clear about what you mean and try to avoid different terms for the same thing.

As suggested, we have standardized the terms used in the text referring to “attendance” to avoid different terms for the same thing.

- At the bottom of page 8, you report the results on breast cancer screening behaviour. I wonder, again, whether or not you excluded women outside the target age range for organised screening on Campania when you look at reasons for reasons to not be screened. Not having been invited seems to be most relevant. However, of course only for women 45-69. Potential (organisational, financial) barriers towards a complete invitational coverage of eligible women should definitely be included in the discussion of this paper!

In response to the point regarding results on breast cancer screening, as we have already reported, our results are referred only to eligible women 45-69 years.

[Line 468-473] Moreover, we agree that organizational barriers may play a role in incomplete invitation of eligible women and indeed data on invitation coverage in Italy show lower rates in Southern Italy (Osservatorio Nazionale Screening. Report 2019.). As suggested, we have included comments on this issue in the discussion section.

- Source of information, section 3.7: Without the complete questionnaire at hand, it is impossible to know what the women could choose from in term of source of information about cancer prevention. Women invited to screening should have received information about not only the screening procedure but also about the usefulness, effectiveness, harms and benefits of screening. Is that sufficient information?

As suggested, we have included the questions of the questionnaire. However, questions on sources of information included only the type of source for cancer prevention such as mass-media, Internet and physicians, eventual need of additional information, and, in this case, the preferred source. These variables have already been included in the results section and discussed in the discussion section.

Discussion

Needs editing: first word out of context. Generally, sentences are too long and winded and could easily be separated in order to increase readability, e.g. first sentence, split after “organized screening programs, …”

As suggested, we have eliminated the typo “Authors” and have reduced the length and edited the first sentence.

4.1. Starts with a statement regarding knowledge on cancer screening. I suggest adding a brief review of relevant research (results) to increase context.

[Lines 366-3374] As suggested, we have discussed knowledge on cancer screening in the context of the literature available on this topic as follows: “likewise, in a previous study, only 0.9% individuals had accurate knowledge about screening programmes, while 6.9% of participants were able to identify the aim of the screening programmes. The majority (73.0%) had accurate knowledge of the goal of two of the screening programmes, while 12.2% of the sample were able to correctly identify the purpose of one of the screening programmes [17]. Moreover, in a study conducted among European women, 26.5% of women correctly knew benefits and harms of mammography, with the lowest proportion among women in Italy (13.3%). Furthermore, 50.9% of Italian women incorrectly believed that mammography screening prevents breast cancer [18]. These findings underline the need for a stronger commitment to the improvement of population education on cancer prevention and specifically on benefits and harms of cancer screening.”

Last paragraph of page 14, “furtherly confirmed” should be edited.

As suggested we have modified “furtherly confirmed” in “confirmed”

First paragraph page 15, edit sentence starting with “As expected…”It is too long and the reader loses the meaning of the results. Also, re-write “higher knowledge” into “accurate knowledge”.

In response to this point, this paragraph discussed the results of the model on attitudes, that has now been deleted according to suggestions of Reviewer 1.

4.2. Do you also consider the possibility that women are overscreened in terms of more often than the recommended 24 months interval (for breast cancer)? Without the questionnaire at hand, I do not know the exact question you asked and whether there is the possibility of a sub-group of women to have been invited to the screening program PLUS attending opportunistic screening. There are two ends from the “recommended interval”.
Please edit and shorten the sentence starting with “According to these results,…”. The information gets lost.

In response to this point, we agree that overscreening may be a possibility. However, as you can see from the questionnaire, we asked women to indicate time from last screening test, and this allowed us to distinguish two groups: those who performed the test within and outside the recommended interval. Thanks to your question, we have now included this issue as a limitation of our study.

As suggested, we have edited and shortened the sentence starting with “According to these results,..”.

I suggest including different publications to refer to participation rate differences across Europe, e.g. the European screening report (Basu P et al., Int J Cancer 2018) and Zielonke et.al. (Int J Cancer, 2020), who also collected examination coverage of organised as well as opportunistic breast cancer screening.

[Lines 424-428] As suggested, we have added publications referred to participation rate differences across Europe, and discussed them as follows: “In particular, the examination of coverage for breast cancer screening in Europe ranges from 49% (East), 62% (West), 64% (North) to 69% (South) [28]. Moreover, Basu et al. reported that overall, only 5.8% and 11.9% of eligible European women have access to breast and cervical cancer screening, respectively [30].”

  1. Basu P, Ponti A, Anttila A, Ronco G, Senore C, Vale DB, Segnan N, Tomatis M, Soerjomataram I, Primic Žakelj M, Dillner J, Elfström KM, Lönnberg S, Sankaranarayanan R. Status of implementation and organization of cancer screening in The European Union Member States-Summary results from the second European screening report. Int J Cancer. 2018;142:44-56. doi: 10.1002/ijc.31043.
  2. Zielonke, N.; Gini, A.; Jansen, E.E.L.; Anttila, A.; Segnan, N.; Ponti, A.; et al. Evidence for reducing cancer-specific mortality due to screening for breast cancer in Europe: A systematic review. Eur. J. Cancer 2020, 127, 191-206. doi: 10.1016/j.ejca.2019.12.010.

In the last paragraph of page 15, you start using different terms for the same thing, although there clearly are differences between “inequality” and “inequity”. Make sure you use these terms coherently!

In response to this point, we were dealing with inequalities and have now accordingly modified the text.

In Table 2 you present that a shockingly high 14.2% of respondent claim to have never heard about cancer screening. THAT is a result to hand over to the responsible stakeholders! Which leads me to the next comment: which stakeholders other than GP are responsible for a successful screening program? The paper would really benefits if the scope of the discussion would be widened to give the reader more context.

[Lines 360-365] As suggested, we have discussed this topic in the Discussion as follows: “Moreover, it is unacceptable that 14.2% of respondents declared to have never heard about cancer screening; this finding reveals that although organized programs have been implemented more than 20 years ago, communication and education on their role have not yet reached a consistent portion of the eligible population, demanding the implementation of more effective strategies to inform and motivate women to attend these consolidated preventive programs.”

This study focused on socio-economic inequality in the use of cancer testing. However, inequality can be present in each phase along the cancer care pathway, so inequalities in screening findings, survival, and in the access to cancer treatment could also be taken into account. Furthermore, deprived women with screen-detected breast cancer are more likely to face barriers like travel time and distance, inability to take time off work, and lack of health information (Bigby J, Cancer Causes Control 2005; Bradley CJ et al., J Natl Cancer Inst 2002). An Italian study found that the breast cancer screening programme reduced disparities in the access to treatment (Zengarini N, Eur J Cancer Prev 2016). I suggest extending the discussion.

As suggested, we have extended the Discussion section about socio-economic inequality in the use of cancer. Moreover, as suggested, we have cited appropriate literature:

  1. Bigby J, Holmes MD. Disparities across the breast cancer continuum. Cancer Causes Control. 2005 Feb;16:35-44. doi: 10.1007/s10552-004-1263-1.
  2. Bradley CJ, Given CW, Roberts C. Race, socioeconomic status, and breast cancer treatment and survival. J Natl Cancer Inst. 2002;94:490-496. doi: 10.1093/jnci/94.7.490.
  3. Zengarini N, Ponti A, Tomatis M, Casella D, Giordano L, Mano MP, Segnan N, Whitehead M, Costa G, Spadea T. Absence of socioeconomic inequalities in access to good-quality breast cancer treatment within a population-wide screening programme in Turin (Italy). Eur J Cancer Prev. 2016;25:538-546. doi: 10.1097/CEJ.0000000000000211.

Conclusions

Edit first line “unacceptably low”

As suggested, in Conclusion section we have edited “unacceptable low” in “unacceptably low”.

Round 2

Reviewer 1 Report

None.

Author Response

My colleagues and I are most grateful for the positive tone of the comment.

Reviewer 2 Report

Thank you for integrating my remarks and comments, well done!

Some last things:

Paragraph starting in line 259 needs editing: commas are missing.

At the end of sentence, line 278, replace comma by full point.

The numbers mentions in line 339 ff are not from Zielonke, N.; Gini, A.; Jansen, E.E.L.; Anttila, A.; Segnan, N.; Ponti, A.; et al. Evidence for reducing cancer-specific mortality due to screening for breast cancer in Europe: A systematic review. Eur. J. Cancer 2020, 127, 191-206. 

but from 

Zielonke N, Kregting LM, Heijnsdijk EAM et al. The potential of breast cancer screening in Europe. Int J Cancer 2020.

line 385: suggestion to increase clarity:
However, these figures are relatively optimistic compared to those presented in the 2019 Report of the Italian National Observatory on screening. Here, a coverage of invitation in Southern Italy 
of 59% and 74% for breast and cervical cancer screening, respectively [10]. Thus, 41% and 26% of eligible women were not invited to screening.

Author Response

Reviewer 2

Paragraph starting in line 259 needs editing: commas are missing.

As suggested, commas have been added.

At the end of sentence, line 278, replace comma by full point.

As suggested a full point has replaced comma.

The numbers mentions in line 339 ff are not from Zielonke, N.; Gini, A.; Jansen, E.E.L.; Anttila, A.; Segnan, N.; Ponti, A.; et al. Evidence for reducing cancer-specific mortality due to screening for breast cancer in Europe: A systematic review. Eur. J. Cancer 2020, 127, 191-206. 

but from 

Zielonke N, Kregting LM, Heijnsdijk EAM et al. The potential of breast cancer screening in Europe. Int J Cancer 2020.

As suggested “Zielonke N, Kregting LM, Heijnsdijk EAM et al. The potential of breast cancer screening in Europe. Int J Cancer 2020.” has been included and the subsequent references have been corrected.

line 385: suggestion to increase clarity:

However, these figures are relatively optimistic compared to those presented in the 2019 Report of the Italian National Observatory on screening. Here, a coverage of invitation in Southern Italy of 59% and 74% for breast and cervical cancer screening, respectively [10]. Thus, 41% and 26% of eligible women were not invited to screening.

This sentence has been changed according to your suggestions.